# Mutational Disruption of TP53: A Structural Approach to Understanding Chemoresistance

**DOI:** 10.3390/ijms26189135

**Published:** 2025-09-18

**Authors:** Ali F. Alsulami

**Affiliations:** Department of Biochemistry, Faculty of Science, King Abdulaziz University, Jeddah 21589, Saudi Arabia; afmalsulami1@kau.edu.sa

**Keywords:** TP53 3D structure mapping, *TP53* mutations, mutation effect prediction, COSMIC mutation analysis

## Abstract

The tumour suppressor protein p53 plays a central role in safeguarding genomic integrity through the regulation of DNA repair, cell cycle arrest, and apoptosis. Mutations in *TP53*, particularly within its DNA-binding domain, are among the most frequent genetic alterations in human cancers and are strongly associated with chemoresistance and poor prognosis. In this study, all TP53 mutations reported in the COSMIC database were systematically mapped onto all experimentally resolved TP53 three-dimensional structures available in the Protein Data Bank, supplemented with AlphaFold-predicted models to achieve full structural coverage. Mutations were classified according to their structural context—protein core, interface regions, ligand- and zinc-binding sites, and intrinsically disordered regions—and evaluated using complementary sequence- and structure-based predictive tools. The analysis revealed distinct mutational hotspots, differential distribution across structural regions, and context-dependent effects on stability and DNA-binding capacity. Notably, a subset of mutations exhibited consistent predictions of high destabilisation across all structural contexts, underscoring their potential as drivers of functional inactivation. By providing a comprehensive structural map of TP53 alterations, this work offers a valuable resource for understanding mutation-specific mechanisms of p53 dysfunction and for guiding the development of precision therapeutic strategies aimed at restoring its tumour-suppressive functions.

## 1. Introduction

The *TP53* gene encodes the tumour suppressor protein *p53*, a vital regulator of cellular homeostasis and genomic integrity [1,2]. *p53* is activated in response to various cellular stresses, including DNA damage, hypoxia, and oncogene activation [3]. Once activated, *p53* triggers a range of protective responses, including cell cycle arrest, apoptosis, senescence, DNA repair, and metabolic reprogramming, all aimed at preventing the propagation of damaged cells. Through these mechanisms, *TP53* plays a key role in maintaining genomic stability and preventing tumour development [4,5].

In normal cells, *p53* activity is tightly regulated by its negative regulator, *MDM2*, which targets *p53* for ubiquitination and subsequent proteasomal degradation [6,7]. However, in many cancers, this balance is disturbed [8,9]. TP53 is the most frequently mutated gene in human cancers, with over 50% of all tumours harbouring alterations in this gene [10,11,12]. Most mutations are missense substitutions within the DNA-binding domain, resulting in the loss of wild-type tumour-suppressive function or the gain of function properties that contribute to tumour progression, metastasis, and therapeutic resistance [13,14].

Given its vital role in cellular defence mechanisms, studying *TP53* mutations offers crucial insights into cancer biology and treatment outcomes [15]. Structural approaches, particularly those combining experimentally determined *p53* conformations and mutation mapping, provide a robust framework for understanding how specific alterations affect protein function [16,17,18]. By linking mutation sites with DNA-binding interfaces, protein–protein interaction surfaces, and core structural elements, researchers can better interpret the functional impacts of *TP53* mutations and their involvement in chemoresistance [19,20,21]. Structurally, missense mutations in *TP53* can have diverse effects, including protein destabilisation, disruption of DNA-binding surfaces, and altered interactions with regulatory proteins. Such structural alterations often lead to loss of function or, in some cases, gain of oncogenic activity, ultimately promoting tumour progression. Importantly, these conformational changes can also affect therapeutic responses, as destabilised or misfolded *p53* variants may fail to activate apoptosis in response to DNA-damaging agents. Consequently, *TP53* structural alterations are increasingly recognised as a major contributor to chemoresistance across multiple cancer types [22,23]. Such structural insights are vital for developing personalised therapeutic strategies, including targeted drug design and mutation-specific interventions, ultimately enhancing treatment effectiveness in *TP53*-mutant cancers [24].

The *p53* protein consists of several functionally distinct domains that are essential for its tumour suppressor activity (Figure 1A). At the N-terminus, it features the transactivation domain (TAD), which interacts with the transcriptional machinery and regulatory proteins such as *MDM2*, *FOXO4* [25]. This is followed by a proline-rich domain involved in apoptotic signalling [26]. The central DNA-binding domain (DBD) is highly conserved and responsible for sequence-specific binding to target gene promoters; most cancer-related mutations occur in this region (Figure 1B). The C-terminal region includes the oligomerisation domain (OD), which allows p53 to form active tetramers, and the regulatory domain, which modulates DNA binding and post-translational modifications [27]. These domains work together to coordinate *p53’s* transcriptional and tumour-suppressive functions [28].

Common *TP53* mutations linked to drug resistance usually occur in the DNA-binding domain, impairing p53’s capacity to regulate genes involved in apoptosis and cell cycle arrest [29]. Hotspot mutations such as R175H, R248Q, R273H, and R282W interfere with DNA contact or protein structure, resulting in the loss of tumour suppressor function and, in some cases, gain-of-function activities that support survival and chemoresistance [30,31]. These mutations can diminish the effectiveness of DNA-damaging agents, such as cisplatin or doxorubicin, by blocking apoptosis [32].

While previous studies have extensively characterised *TP53*-specific mutations in terms of their frequency, functional impact, and association with clinical outcomes [33,34,35,36,37], fewer have integrated this information with high-resolution structural data to contextualise mutations at the atomic level. By complementing existing genomic and functional analyses with a structure-based framework, this study provides a more comprehensive understanding of how specific mutations may alter key functional domains of *p53*, such as DNA binding, dimerisation, and protein–protein interaction interfaces and contribute to chemoresistance. Integrating both sequence-based and structure-based predictive tools enhances the interpretability of *TP53* mutations in a therapeutic context.

## 2. Results

A comprehensive set of 277 experimentally determined 3D structures of *TP53* was obtained from the Protein Data Bank (PDB) [38]. These structures encompass mutations at 368 of the 393 amino acid residues, representing approximately 93.6% of the protein sequence. Mutation data from the COSMIC database [39] identified a total of 1472 unique variants, with 340 unique mutations (23%) located in predicted intrinsically disordered regions, as identified by DISOPRED3 [40], which estimates that 36.5% of the TP53 sequence is disordered (Figure 1A). Importantly, no full-length 3D structure of *TP53* has been solved; existing structures generally represent isolated domains or short peptides, often in complex with other molecules. To attain complete structural coverage, AlphaFold-predicted models [41] were integrated to fill in regions not covered by experimental data.

A total of 1472 unique *TP53* variants were mapped onto 277 experimentally resolved 3D structures, accounting for 185,383 individual missense mutation events (Appendix A). These Missense mutations were categorised based on their structural context, including their location in the protein core, at interface regions, in non-interface surface areas, near ligand-binding sites, within protein–protein interaction surfaces, or around the zinc-binding site. Additionally, the functional impact of each mutation was assessed using a combination of sequence-based and structure-based predictive methods.

Analysis of the AlphaFold-predicted monomeric structure of *TP53* (Figure 1A), which covers the entire sequence (residues 1–393) and lacks any bound cofactors, ligands, or DNA, offers a comprehensive overview of the mutational landscape. The predicted AlphaFold model closely aligns with the experimental structure of the DNA-binding domain (PDB ID: 8DC8), showing an RMSD of 0.345, which reflects a high structural similarity. Based on this model, 54.3% of the mutations are situated at protein interfaces, while 21.4% occur within the protein core, and 24.3% are located on non-interface surface regions (Appendix A).

The structural and functional impact of *TP53* mutations within the DNA-binding domain was examined across three structural contexts: (i) monomeric forms bound to cofactors and ligands, (ii) hetero-oligomeric complexes, and (iii) homo-oligomeric assemblies in complex with DNA. A detailed analysis of experimental structure PDB ID 8DC8 with (Y220C) mutant [42], which represents the monomeric DNA-binding domain of TP53 (residues 91–289) bound to a zinc ion (Zn^2+^) and (R50) ligand, shows that 988 out of the 1472 reported mutations are located within this region. Of these, 3.1% occur near the zinc-binding site, 8.7% around the ligand-binding site, 33.9% are found within the core region, 18.8% are located at the interface, and 35.4% in the non-interface region (Figure 2C and Figure 3C).

An alternative structural form of this domain is represented by the Hetero 6-mer PDB structure PDB ID 8R1G [43], which facilitates further classification of the same 1060 mutations within a multimeric context. In this assembly, 19.4% of the mutations occur at dimer interfaces, 5.2% near zinc-binding interactions, 35.3% in the core, 26.7% in non-interface regions, and 13.4% at protein interfaces (Figure 2A and Figure 3A).

Furthermore, analysis of the Homo 4-mer DNA-bound *TP53* structure PDB ID: 2ATA, 7EEU [44] with R282W mutant provides insights into mutations at the protein–DNA interface. In this structure, 9.2% of mutations are located at the DNA-binding interface. Additionally, 4.1% of mutations are found near the zinc-binding site, 16.9% occur between protein dimers, and 36.7% reside within the protein core, representing the largest proportion (Figure 2B and Figure 3B,D).

Comparing experimentally solved wild-type and mutant TP53 structures for the most frequently altered residues (Table 1), several distinct structural and functional consequences become evident. The R249S mutation appears to stabilise its surrounding region, enabling binding in higher-order assemblies such as those seen in PDB ID: 7EEU. Structural alignment of PDB ID: 3D09 (R249S mutant) with PDB ID: 7EEU suggests that without this stabilising effect, inter-domain interactions would not be maintained. In wild-type *p53*, R249 stabilises the core domain by linking loops L2 and L3 through a bidentate salt bridge and hydrogen bonds; substitution with serine disrupts these interactions, weakening structural integrity and reducing DNA-binding affinity. The zinc-binding site in the TP53 core domain is critical for maintaining correct folding of the DNA-binding surface by coordinating a zinc ion with specific cysteine and histidine residues. The G245S mutation, located near this site, can disturb local structure and indirectly impair zinc coordination, destabilising the DNA-binding region and reducing target-sequence recognition. While R175 is not a zinc-coordinating residue, it stabilises the loop–sheet–helix motif that positions the zinc ion and DNA-contacting loops, forming a salt bridge with D184 to maintain zinc-pocket geometry (Figure 2A). The Y220C mutation generates a unique surface cavity at the ligand site, offering a promising target for precision therapeutics (Figure 2C). Mutations at the protein–DNA interface, such as R273C, markedly reduce DNA-binding affinity despite retaining near wild-type structural stability; similarly, the R248Q mutation disrupts key arginine–phosphate backbone contacts, severely compromising DNA binding while preserving overall fold (Figure 2B).

Most *TP53* mutations occur within the DNA-binding domain (DBD), with 1047 mutations, followed by the proline-rich region, which contains 86 mutations (Figure 1C). Based on mutation frequency, *TP53* variants are categorised into four groups

Very high-frequency mutations (>500 occurrences):

A total of nine mutations are in this category, all located within the DNA-binding domain. Among them, four occur at the protein–DNA interface, one is near the ligand- and zinc-binding sites, and two are in non-interface regions (Table 1).

High-frequency mutations (100–500 occurrences):

This group includes 66 mutations, with 65 located in the DNA-binding domain and one mutant (P72R) located in the proline-rich region.

Moderately high-frequency mutations (50–100 occurrences):

Comprising 62 mutations, all are located within the DNA-binding domain.

Lower high-frequency mutations (20–50 occurrences):

There are 170 mutations in this range. Most are within the DNA-binding domain, except for (P72A) in the proline-rich region, (V31I) in the transactivation domain, and G334V in the Tetramerisation domain.

Mutation effects were predicted using structure-based approaches, mCSM [45], Maestro [46], and FoldX [47], alongside the sequence-based method AlphaMissense [48]. For high-frequency mutations, predictions from FoldX showed a strong correlation with AlphaMissense, except for variants located at the protein–DNA interface, where greater variability between tools was observed. All predictive methods consistently classified Y220C, located at the ligand-binding site, as highly destabilising and pathogenic (Table 1). In contrast, R249S yielded a borderline stability change in mCSM but was predicted by Maestro and FoldX to have stronger destabilising effects, a result that was in agreement with the high pathogenicity score assigned by AlphaMissense.

Across the three TP53 structural contexts analysed—monomeric (PDB 8DC8, 988 mutations), DNA-bound tetramer (PDB 7EEU, 1045 mutations), and heterohexameric assembly (PDB 8R1G, 1060 mutations)—complete agreement among the three structure-based prediction methods was observed for a subset of mutations. Specifically, agreement occurred for 326 mutations (33%) in 8DC8, 356 mutations (34%) in 7EEU, and 331 mutations (31%) in 8R1G. Within these consensus sets, the majority of mutations were classified as moderately destabilising, a smaller portion as highly destabilising, and only a minority as moderately stabilising. For the remaining mutations, predictions varied among the tools, indicating inconsistent assessments of their structural impact. On the other hand, analysis using the sequence-based method for 1467 unique mutations predicted 111 mutations (7.6%) as ambiguous, 572 mutations (39%) as likely benign, and 784 mutations (53.4%) as likely pathogenic (Appendix A). The intrinsically disordered regions (IDRs) of TP53, spanning residues 1–96, 290–322, and 358–392, encompass a total of 340 mutations. Structure-based prediction tools showed only 29% agreement in classifying these mutations as moderately destabilising. In contrast, the sequence-based method predicted 13 mutations (3.8%) as ambiguous, 15 mutations (4.4%) as likely pathogenic, and the vast majority—312 mutations (91.8%)—as likely benign.

In the DNA-binding domain (PDB ID: 7EEU), there was 100% agreement between the sequence-based method and all three structure-based tools for the 82 mutations classified as highly destabilising. In the heterohexameric assemblies (PDB IDs: 8R1G and 8DC8), 97.8% of the mutations classified as highly destabilising by the structure-based tools were assigned the same classification by the sequence-based method (Figure 4).

Among the 82 mutations predicted to be highly destabilising in PDB ID: 7EEU, the vast majority (78 mutations) are located within the protein core. C238S is situated at the zinc-binding site, R267G occurs at the dimer interface, and only three mutations (V274A/D/G) are found at the protein–DNA interface.

In PDB ID: 8DC8, six mutations (I195K, Y234R, L145Q, L257P, L257Q) occur near the ligand-binding site, three (C238S, C238R, C238G) are around the zinc ions, seven are in non-interface residues, and 66 mutations lie within the core region (Appendix A).

For the high-order assembly (PDB ID: 8R1G), two mutations (C238S/G) are at the zinc-binding site, eight are located at dimer interfaces (Y126H/D/N/S, R282G, L130H, F113S, W146G), 77 mutations occur in the core, and six occur in non-interface residues (Appendix A).

Across all three structures, three unfrquent mutations (V157E, Y234R, I195K) consistently appear in the core region and are predicted as highly destabilising by all methods, suggesting a substantial impact on TP53 stability and binding.

## 3. Discussion

The comprehensive structural mapping of *TP53* mutations onto both experimentally determined and AlphaFold-predicted models provides mechanistic insights into how alterations in p53 compromise its tumour-suppressive functions and contribute to chemoresistance [49]. The predominance of mutations within the DNA-binding domain (DBD) underscores its central role in regulating transcriptional programmes critical for cell-cycle arrest, apoptosis, and DNA repair. Structural analysis reveals that these mutations exert their effects through two primary mechanisms: direct disruption of DNA contact and indirect destabilisation of the domain architecture, which can impair long-range allosteric communication with other functional regions.

Hotspot mutations such as R248Q and R273C, which reside at the protein–DNA interface, primarily reduce p53’s DNA-binding affinity without substantially altering global stability. As a result, mutant p53 fails to activate transcription of pro-apoptotic genes like *BAX* and *PUMA*, rendering cells less responsive to DNA-damaging chemotherapeutics, including cisplatin and doxorubicin. In contrast, structural core mutations such as V157E, Y234R, and I195K cause profound destabilisation of the *p53* fold, as consistently predicted by multiple structure- and sequence-based tools. These alterations compromise protein stability, promote misfolding, and frequently result in dominant-negative effects, further exacerbating functional loss in heterozygous tumours.

Mutations affecting metal coordination and ligand-binding sites, including R175H, C238S/G/R, G245S, and Y220C, demonstrate how subtle perturbations can lead to chemoresistance through multiple mechanisms. Disruption of zinc coordination impairs proper folding of the DBD and reduces transcriptional activity, while the Y220C substitution generates a solvent-accessible cavity, destabilising the domain but simultaneously creating a druggable pocket. This feature has already been exploited by small-molecule stabilisers that restore wild-type-like function, illustrating the therapeutic potential of structure-guided drug design.

Furthermore, the analysis highlights the functional importance of *p53* oligomerisation. Mutations at the dimer interface, such as R282W and L130H, destabilise tetramer formation, a prerequisite for DNA binding and transactivation. Loss of tetrameric *p53* diminishes the transcriptional activation of genes involved in apoptosis and cell-cycle control, directly linking structural disruption to therapy resistance.

Interestingly, mutations located within intrinsically disordered regions (IDRs) exhibit a different pattern, as these segments demonstrate higher tolerance to substitutions due to their structural flexibility. However, mutations within IDRs may still perturb regulatory interactions with partners such as *MDM2* and transcriptional co-activators, suggesting that their role in chemoresistance might be more subtle and context-dependent.

The use of AlphaFold’s artificial intelligence-based structure prediction played a critical role in enabling full structural coverage of *TP53* mutations. This capability allowed us to systematically analyse mutation sites that were previously difficult to characterise, particularly those located in disordered regions or complex interfaces. By overcoming limitations of experimental structural methods, AlphaFold provides deeper insights into the mechanisms by which *TP53* mutations disrupt protein function. This methodological advancement also highlights future directions for integrating computational and experimental tools to facilitate drug discovery and personalised therapeutic strategies targeting structurally vulnerable TP53 mutants.

Comparative assessment of predictive tools (mCSM, FoldX (V.5), Maestro (1.1), and AlphaMissense (V.0.0.1)) revealed strong concordance for core and ligand-binding mutations, supporting their robust destabilising effects. Discrepancies observed for DNA-contacting residues and disordered regions highlight the limitations of individual methods and underscore the need for integrating multiple predictive strategies with high-resolution structural mapping.

Overall, structural analysis of *TP53* mutations highlights several regions and residues that are particularly critical for protein stability, DNA binding, and protein–protein interactions. Among these, mutations within the DNA-binding domain (residues 94–292)—particularly R175H, R248Q/W, R273H, and Y220C—are associated with profound destabilisation, impaired transcriptional regulation, and increased chemoresistance. Notably, the Y220C mutation creates a solvent-accessible cavity that has been successfully targeted in previous studies using small molecules to restore TP53 stability and function, demonstrating its high druggability potential. Similarly, interface-disrupting mutations such as R337H within the tetramerisation domain suggest opportunities for structure-guided drug design aimed at stabilising TP53 oligomerisation.

These findings provide a structural framework for precision oncology by linking mutation hotspots to potential therapeutic strategies. Future research should focus on:Structure-based drug discovery targeting druggable cavities like Y220C and other destabilised cores.Computational–experimental pipelines to screen and validate small molecules that restore TP53 folding and DNA-binding capacity.

As this is a purely computational study, the findings presented here are based on structural predictions and require experimental validation. While the computational analyses provide valuable insights into the potential impact of TP53 mutations, future studies integrating biochemical assays, structural characterisation, and drug-sensitivity experiments will be essential to confirm these predictions. Combining computational and experimental approaches will strengthen the conclusions and enhance the translational relevance of our findings.

## 4. Materials and Methods

UniProt IDs [50] of TP53 genes (P04637) were mapped to corresponding Protein Data Bank (PDB) entries to retrieve all available experimental structures. To ensure consistent residue numbering across datasets, all PDB files were renumbered using PDBrenum [51], enabling precise localisation of mutations on the 3D structures. Structural coverage was then calculated to identify regions with and without experimental data. For structurally unresolved regions, AlphaFold models (v2.0) were incorporated to achieve a complete structural representation of all reported COSMIC mutations.

Mutation data were obtained from the COSMIC database (v102). The dataset was cleaned and curated to remove duplicates and erroneous entries. Mutation frequencies were calculated to assess prevalence. Each mutation was then mapped to its corresponding position in either the experimental PDB structures or AlphaFold models to enable 3D contextualization.

### Solvent Accessibility and Residue Classification

Solvent accessibility was determined using the FreeSASA (2.1.2) tool [52], which was installed and used locally with default parameters. Residues were classified according to exposure levels:Core residues: SASA < 25Non-interface residues: 25 ≤ SASA ≤ 80Interface residues: SASA > 80

This classification offered insight into the spatial environment of each mutated residue. To evaluate proximity to key functional regions, mutations were mapped using multiple distance thresholds [53].

6 Å for ligand-binding sites5 Å for protein-DNA interfaces, zinc ion coordination sites, and homo-/hetero-dimer interfaces

This ensured precise mapping of mutations within relevant structural and functional contexts. To assess the impact of mutations, both structure- and sequence-based tools were utilised. mCSM, FoldX, and Maestro were employed to estimate variations in protein stability and interaction energy (ΔΔG kcal/mol) derived from the 3D structure. Additionally, AlphaMissense, a deep learning-driven predictor, was used to evaluate the pathogenicity of variants based on sequence data. It is important to note that the sign conventions differ between methods. For mCSM and Maestro, negative ΔΔG values indicate destabilising mutations, whereas positive values suggest stabilisation. In contrast, FoldX uses the opposite convention: positive ΔΔG values represent destabilising mutations, and negative values indicate stabilisation. To harmonise the interpretation, we focused on the qualitative classification (“stabilising” vs. “destabilising”) rather than absolute values. Additionally, pathogenicity scores were derived from AlphaMissense, where scores ≥ 0.56 are classified as likely pathogenic and scores < 0.56 as likely benign, based on the recommended threshold from the AlphaMissense model. This integrated analysis facilitated a comprehensive functional interpretation of TP53 mutations.

## 5. Conclusions

In this study, a comprehensive structural characterisation of all *TP53* mutations reported in the COSMIC database was mapped onto all experimentally resolved *TP53* three-dimensional structures. Integrating structural mapping with functional classification, we identified patterns in mutation distribution across core, interface, and DNA-binding regions, as well as their predicted impact on protein stability and interactions. The results highlight recurrent hotspots that coincide with structurally and functionally critical residues, providing insight into the molecular mechanisms underlying *TP53* dysfunction in cancer. This large-scale structural perspective not only refines our understanding of *TP53* pathogenicity but also offers a valuable resource for guiding future experimental studies, functional validation, and the rational design of targeted therapeutic strategies.

## Figures and Tables

**Figure 1 ijms-26-09135-f001:**
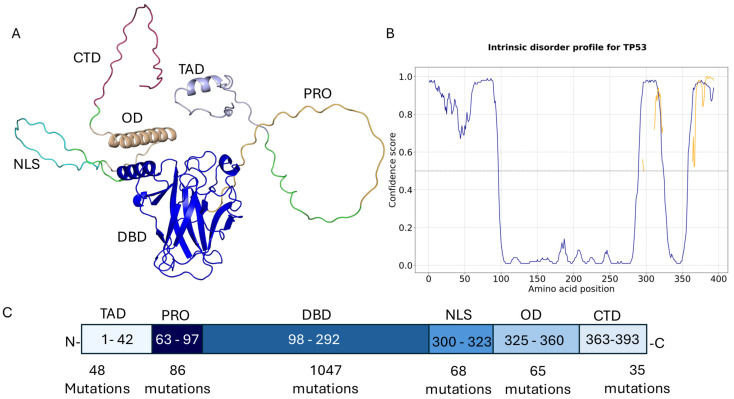
Domain architecture of the *TP53*. (**A**) AlphaFold-predicted structure of full-length *TP53*, with each functional domain highlighted in a distinct colour: transactivation domain 1 (TAD1), transactivation domain 2 (TAD2), proline-rich domain (PRD), DNA-binding domain (DBD), oligomerisation domain (OD), and C-terminal regulatory domain (CTD). (**B**) Predicted intrinsically disordered regions (IDRs) based on DISOPRED3 scores: regions with disorder score > 0.5 are shown as disordered, and those < 0.5 as ordered. (**C**) Schematic representation showing the start and end positions of each domain. The number of reported mutations in each domain, based on COSMIC data, is indicated under the corresponding region.

**Figure 2 ijms-26-09135-f002:**
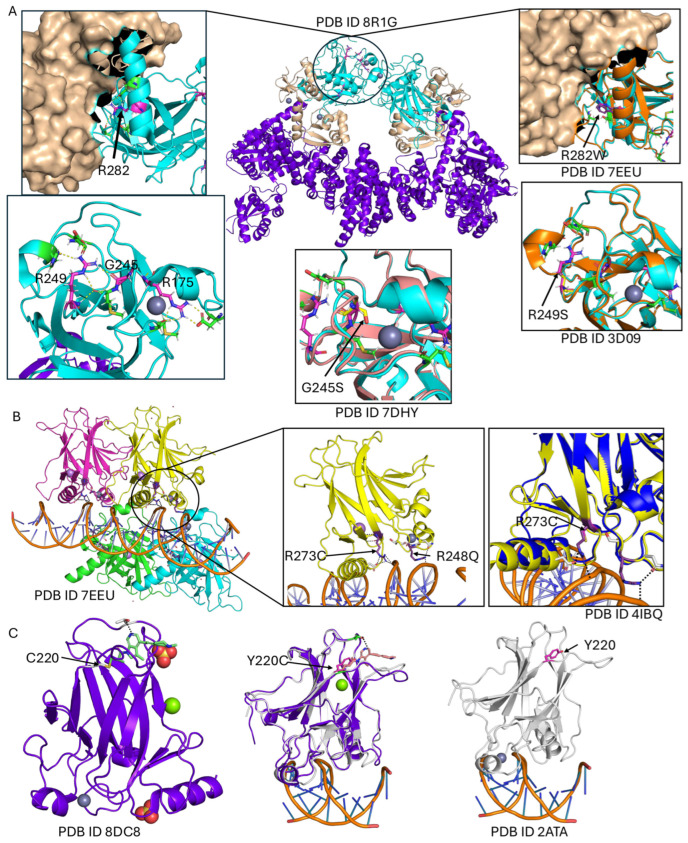
(**A**) The heterohexamer structure of *TP53* is shown in cyan, with the most frequent mutations—R249, G245, R282, and R175—highlighted in magenta sticks. Zoomed-in panels illustrate the structural alignment between the wild-type (cyan) and mutant (orange) proteins. (**B**) The homotetrameric *TP53*–DNA complex is displayed, with each domain shown in a different colour. The most frequent mutation occurs at the protein–DNA interface, highlighted in purple sticks. The PDB structure 4IBQ is used, with the mutant depicted in blue. Structural alignment highlights the R273C mutant in magenta sticks. (**C**) The TP53 wild-type structure (PDB 8DC8) is shown in purple-blue, with the ligand represented in green sticks and residue C220 in white sticks. The mutant TP53 structure is coloured white, with the mutation at the ligand-binding site shown in magenta sticks.

**Figure 3 ijms-26-09135-f003:**
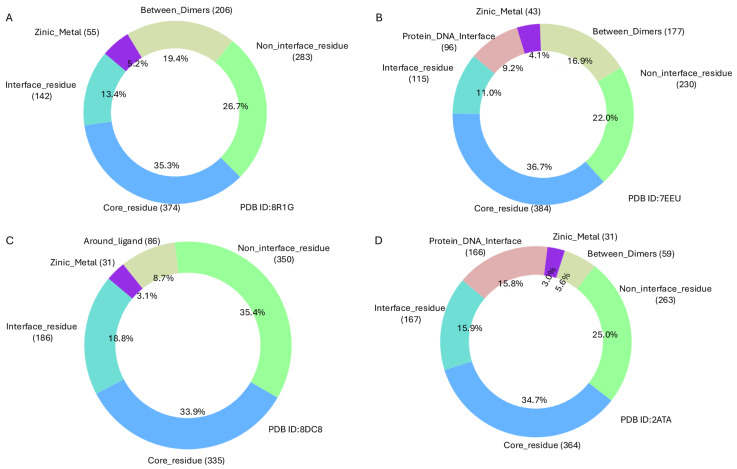
Annotation and classification of mutations mapped onto the protein structure for the selected PDB entry.

**Figure 4 ijms-26-09135-f004:**
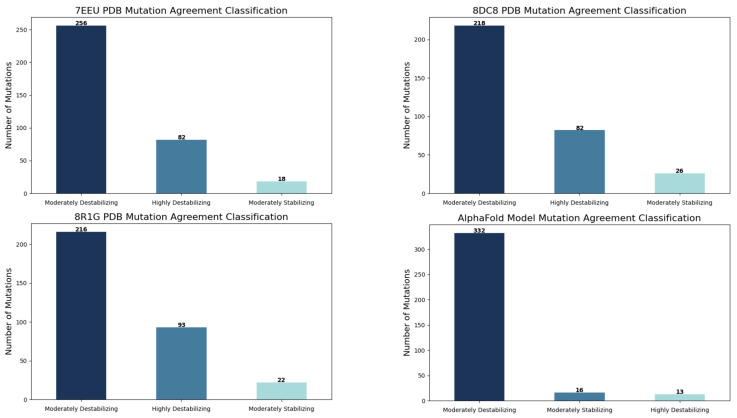
Agreement between structural-based and sequence-based prediction tools for mutations mapped on selected PDB structures. For experimental structures, moderately stabilising mutations showed the highest agreement, whereas moderately destabilising mutations showed the lowest agreement.

**Table 1 ijms-26-09135-t001:** High-Frequency *TP53* Mutations Occurring More Than 500 Times, with Structural Context Classification.

Mutations	Frequency	PDB	Locations	mCSM	Maestro	FoldX	AlphaMissense
R175H	2513	8R1G	around ZN metal	−1.07	0.3237	11.13	0.98
R248Q	1692	7EEU	protein DNA interface	−0.85	−0.24	−0.98	0.99
R273H	1521	7EEU	protein DNA interface	−1.96	0.44	1.22	0.98
R273C	1492	7EEU	protein DNA interface	−2.02	−0.11	1.32	0.99
R248W	1308	7EEU	protein DNA interface	−1.277	−0.11	0.01	0.99
R282W	1165	8R1G	between dimers interface	−1.41	1.02	9.07	0.94
G245S	809	8R1G	around ZN metal	−1.536	0.69	4.03	0.97
Y220C	761	2ATA	ligand binding site	−3.524	3.27	5.11	0.97
R249S	646	8R1G	non-interface	−1.859	1.85	2.72	0.99

## Data Availability

The original contributions presented in this study are included in the article/Appendix A. Further inquiries can be directed to the corresponding author.

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
