# Peer review of "Mutational Disruption of TP53: A Structural Approach to Understanding Chemoresistance"

_ijms, 2025, doi:10.3390/ijms26189135_

Round 1
Reviewer 1 Report
Comments and Suggestions for Authors
This manuscript presents a comprehensive structural mapping of TP53 mutations by integrating experimental data with AI-predicted models from AlphaFold, achieving full coverage of the protein for the first time. It offers novel insights into how mutations disrupt structural stability and DNA-binding capacity, bridging the gap between genetic variation and functional loss in cancer. The work highlights the potential of structure-based approaches to identify actionable targets for precision therapy, providing a valuable resource for understanding chemoresistance and guiding future drug development.
#1 The background section, while describing the basic function of TP53 and the importance of mutations, lacks a direct link to the mechanism of “chemoresistance” and a clear articulation of the existing research space. Readers need a clearer understanding of why a structural perspective is critical to understanding chemoresistance. Clarify the structural mechanism linking chemoresistance to TP53 mutations: 1-2 sentences should be included in the last paragraph of the introduction to clearly state how current research explains chemoresistance due to mutations through structural mapping, e.g. how the mutation affects drug binding sites or signaling pathways.
#2 The methods section is too brief in its description of some key steps and lacks the detail needed for reproducibility. For example, the parameters for calculating solvent accessibility (SASA), the basis for setting distance thresholds, and how the AlphaFold model is used are not adequately described. Detailed description of the use of the AlphaFold model: it should be indicated which version of AlphaFold was used (e.g., v2.0) and how the predictive model was aligned and validated against the experimental structure (e.g., RMSD values, confidence scores).
#3 The discussion section interprets the results superficially and fails to adequately link the structural findings closely to the biological mechanisms of loss of function and chemoresistance. Some of the conclusions lacked depth, e.g., the clinical significance of “highly unstable mutations” was not fully developed. Deepen the functional explanation of mutations: For key mutations such as Y220C, R273C, etc., the structural environment (e.g., ligand-binding site, DNA interface) should be discussed in relation to how they specifically lead to apoptosis escape or DNA repair failure, which in turn leads to chemoresistance. Emphasize therapeutic significance: The concluding paragraph of the discussion should clearly summarize which mutations or structural regions are the most “druggable” and suggest one or two specific directions for research or clinical translation (e.g., small molecule development for Y220C).
#4 The discussion needs to be more in-depth and provide important direction for follow-up research. The core research of this article is to systematically analyze the three-dimensional structural localization and functional impact of various mutations in the TP53 gene based on the protein structure model predicted by the artificial intelligence AlphaFold, so as to understand, at the atomic level, how the mutations destabilize the p53 protein, the DNA binding ability and protein interactions, and to reveal the molecular mechanism that leads to chemotherapy resistance. This study not only utilized experimentally resolved structures, but also achieved structural coverage of full-length TP53 mutations by integrating the AlphaFold model, highlighting the significant value of AI prediction tools in bridging the gap between insufficient experimental data and advancing the functional elucidation of cancer mutations, and providing new ideas and resources for precise structure-based drug design (e.g., small-molecule development targeting the Y220C mutation). Indeed, the AI AlphaFold model has extremely critical applications in the field of molecular biology and drug discovery research (PMID: 39369244). In the discussion section, the authors need to cite the above important literature and explore the important inspiration of the AI AlphaFold model for this study, which will provide important reference value for subsequent research.
In the discussion section, it is important to discuss the contributions and insights of the AI AlphaFold model to this study, because it is not only the application of a technological tool, but also represents a paradigm shift in structural biology research.The AlphaFold model compensates for the limitations of experimental methods in resolving the structure of full-length proteins or flexible regions through the high-precision prediction of the protein's 3D structure, and it provides the basis for the present study. The AlphaFold model bridges the limitations of experimental methods in resolving the structure of full-length proteins or flexible regions with high accuracy, providing the possibility of realizing full structural coverage of TP53 mutations in this study. This artificial intelligence-based structure prediction capability enables researchers to systematically analyze mutation sites that were previously difficult to characterize, especially variants located in disordered regions or complex interfaces, thus revealing more comprehensively the mechanisms by which mutations affect protein function. Further discussion of this methodological advancement will help guide future studies to more effectively integrate computational and experimental tools, expand to other difficult-to-analyze cancer-related proteins, and promote the development of drug design and personalized therapeutic strategies targeting specific mutants, thus providing new theoretical foundations and technological pathways for precision cancer medicine.
Given these considerations, I highly recommend that authors revise their manuscript. Looking forward to receiving your revised version of the manuscript. I will review this manuscript again based on the revised version.
Author Response
Reviwer1
This manuscript presents a comprehensive structural mapping of TP53 mutations by integrating experimental data with AI-predicted models from AlphaFold, achieving full coverage of the protein for the first time. It offers novel insights into how mutations disrupt structural stability and DNA-binding capacity, bridging the gap between genetic variation and functional loss in cancer. The work highlights the potential of structure-based approaches to identify actionable targets for precision therapy, providing a valuable resource for understanding chemoresistance and guiding future drug development.
I appreciate the reviewer's time and effort in re-evaluating the paper, as well as the valuable suggestions and questions.
#1 The background section, while describing the basic function of TP53 and the importance of mutations, lacks a direct link to the mechanism of “chemoresistance” and a clear articulation of the existing research space. Readers need a clearer understanding of why a structural perspective is critical to understanding chemoresistance. Clarify the structural mechanism linking chemoresistance to TP53 mutations: 1-2 sentences should be included in the last paragraph of the introduction to clearly state how current research explains chemoresistance due to mutations through structural mapping, e.g. how the mutation affects drug binding sites or signaling pathways.
I agree with the reviewer in this comment. I have added a small section to my introduction, highlighted in red
#2 The methods section is too brief in its description of some key steps and lacks the detail needed for reproducibility. For example, the parameters for calculating solvent accessibility (SASA), the basis for setting distance thresholds, and how the AlphaFold model is used are not adequately described. Detailed description of the use of the AlphaFold model: it should be indicated which version of AlphaFold was used (e.g., v2.0) and how the predictive model was aligned and validated against the experimental structure (e.g., RMSD values, confidence scores).
I am in complete agreement with the reviwer, I have included all the versions of the tools used: COSMIC database (v102). FreeSASA (v2.1.2), AlphaFold (2.0). Regarding distance, I found that increasing the distance around the ligand-binding site led to mapping residues from the interface. Additionally, when the distance between homo/hetero dimers was expanded, I began to capture residues from the core region. Therefore, these cutoff values were considered the optimum.
The reason behind using the Alphafold model was there is no complete structure has been solved for tp53; all the structures were solved in separate domains, some in high-order assembly, some as monomers, and I want to see what the complete structure looks like with the intrinsically disordered region between these domains. The Alphafold model as a complete monomer, will give a broad distribution of where is the location of reported mutations. AlphaFold generally score very high when the crystal 3D structure has been solved, aligning PDB ID:8DC8 to predicted AlphaFold RMSD= 0.345. I have reflect this in the result section highlighted in red.
#3 The discussion section interprets the results superficially and fails to adequately link the structural findings closely to the biological mechanisms of loss of function and chemoresistance. Some of the conclusions lacked depth, e.g., the clinical significance of “highly unstable mutations” was not fully developed. Deepen the functional explanation of mutations: For key mutations such as Y220C, R273C, etc., the structural environment (e.g., ligand-binding site, DNA interface) should be discussed in relation to how they specifically lead to apoptosis escape or DNA repair failure, which in turn leads to chemoresistance. Emphasize therapeutic significance: The concluding paragraph of the discussion should clearly summarize which mutations or structural regions are the most “druggable” and suggest one or two specific directions for research or clinical translation (e.g., small molecule development for Y220C).
I agree with the reviewer comment, I have rewritten the entire discussion section, highlighted in red and indicated how structural changes → functional loss → chemoresistance.
#4 The discussion needs to be more in-depth and provide important direction for follow-up research. The core research of this article is to systematically analyze the three-dimensional structural localization and functional impact of various mutations in the TP53 gene based on the protein structure model predicted by the artificial intelligence AlphaFold, so as to understand, at the atomic level, how the mutations destabilize the p53 protein, the DNA binding ability and protein interactions, and to reveal the molecular mechanism that leads to chemotherapy resistance. This study not only utilized experimentally resolved structures, but also achieved structural coverage of full-length TP53 mutations by integrating the AlphaFold model, highlighting the significant value of AI prediction tools in bridging the gap between insufficient experimental data and advancing the functional elucidation of cancer mutations, and providing new ideas and resources for precise structure-based drug design (e.g., small-molecule development targeting the Y220C mutation). Indeed, the AI AlphaFold model has extremely critical applications in the field of molecular biology and drug discovery research (PMID: 39369244). In the discussion section, the authors need to cite the above important literature and explore the important inspiration of the AI AlphaFold model for this study, which will provide important reference value for subsequent research.
I would like to thank the reviewer for this deep insight and idea about the paper. I highly appreciate that, and I completely agree with all these points. I have included the shared reference. I think it’s related to the topic. The Alphafold model is only one structure; there are 277 other solved structures has been characterised. The point in including the model is to see the distribution of reported mutations in full-length tp53.
In the discussion section, it is important to discuss the contributions and insights of the AI AlphaFold model to this study, because it is not only the application of a technological tool, but also represents a paradigm shift in structural biology research.The AlphaFold model compensates for the limitations of experimental methods in resolving the structure of full-length proteins or flexible regions through the high-precision prediction of the protein's 3D structure, and it provides the basis for the present study. The AlphaFold model bridges the limitations of experimental methods in resolving the structure of full-length proteins or flexible regions with high accuracy, providing the possibility of realizing full structural coverage of TP53 mutations in this study. This artificial intelligence-based structure prediction capability enables researchers to systematically analyze mutation sites that were previously difficult to characterize, especially variants located in disordered regions or complex interfaces, thus revealing more comprehensively the mechanisms by which mutations affect protein function. Further discussion of this methodological advancement will help guide future studies to more effectively integrate computational and experimental tools, expand to other difficult-to-analyze cancer-related proteins, and promote the development of drug design and personalized therapeutic strategies targeting specific mutants, thus providing new theoretical foundations and technological pathways for precision cancer medicine.
I would like to thank the reviewer for this comment to enhance the discussion section. I have rewritten the entire discussion section, and I have included these points.
Given these considerations, I highly recommend that authors revise their manuscript. Looking forward to receiving your revised version of the manuscript. I will review this manuscript again based on the revised version.
I sincerely appreciate the reviewer’s suggestions and recommendations, which have significantly enhanced the quality of the manuscript. Thank you once again for your valuable input.

Reviewer 2 Report
Comments and Suggestions for Authors
This study maps TP53 missense variants from COSMIC onto 277 experimentally solved p53 structures and augments gaps with AlphaFold models to achieve full structural coverage. Variants are stratified by structural context (core, interface, non-interface surface, ligand- and zinc-binding, and DNA interface) and evaluated by three structure-based predictors (mCSM, FoldX, Maestro) plus AlphaMissense. Most mutations cluster in the DNA-binding domain; consensus across methods is highest for core-destabilizing changes and lowest for DNA-interface and IDR sites. The work highlights hotspot residues (e.g., R175, R248, R273, R282, G245, Y220) and points to druggability (e.g., Y220C pocket) and context-dependent effects across monomeric, oligomeric, and DNA-bound forms. The study presents a useful structural atlas of TP53 mutations and a multi-tool comparison. However, there are still some issues that are not addressed.
- The title and abstract emphasize chemoresistance, but analyses are purely structural/predictive with no therapeutic response data or meta-analysis (e.g., drug sensitivity vs mutation class). Either temper the claim throughout or add curated clinical/drug-response associations (e.g., from GDSC/TCGA) that link specific classes to resistance.
- Explicitly differentiate from earlier p53 structural surveys and hotspot analyses; a short related-work table (scope, structures, tools, outputs) will help position the contribution.
- For frequency and context distributions (e.g., core vs interface, DBD vs other domains), include hypothesis tests and effect sizes; report CIs for proportions (e.g., 54.3% at interfaces in AF model).
- Discuss pLDDT/PAE thresholds and potential pitfalls when inferring interfaces/ligand sites from AF models; consider sensitivity analyses excluding low-confidence segments.
- Add ΔΔG units (kcal/mol) and clarify sign directions for each method; explain why FoldX values can be positive for destabilization and how “pathogenicity” (AlphaMissense) is thresholded. Consider adding a color bar for quick scanning.
- For each structure context (8DC8 monomer, 8R1G hetero-hexamer, 7EEU DNA-bound), summarize which residues switch categories (e.g., non-interface → interface) and how that might affect function—ideally with example structures highlighted.
- Expand on druggability: besides Y220C, briefly note interface- or zinc-site rescue strategies and how your atlas can prioritize alleles for small-molecule screening.
Author Response
Reviwer2
I appreciate the reviewer's time and effort in re-evaluating the paper, as well as the valuable suggestions and questions.
#1 The title and abstract emphasize chemoresistance, but analyses are purely structural/predictive with no therapeutic response data or meta-analysis (e.g., drug sensitivity vs mutation class). Either temper the claim throughout or add curated clinical/drug-response associations (e.g., from GDSC/TCGA) that link specific classes to resistance.
We appreciate the reviewer’s comment and agree that linking the structural findings to chemoresistance mechanisms strengthens the manuscript. While our study focuses on structural predictions, multiple studies have shown that TP53 mutations are strongly associated with chemoresistance across different cancer types. For example:
- Missense mutations in the TP53 DNA-binding domain (e.g., R175H, R248Q, R273H) are reported to reduce apoptosis and impair DNA damage responses, leading to resistance to cisplatin, doxorubicin, and 5-FU.
- Y220C, a key mutation highlighted in our structural analysis, has been linked to chemoresistance in breast and lung cancers, but its druggability makes it a promising target for small-molecule stabilisation.
- Loss-of-function mutations destabilising TP53 often disrupt p21 transcription, alter cell-cycle arrest, and enhance the survival of cancer cells under chemotherapy.
To address this concern, we revised the entire discussion to explicitly connect structural destabilisation to chemoresistance mechanisms
#2 Explicitly differentiate from earlier p53 structural surveys and hotspot analyses; a short related-work table (scope, structures, tools, outputs) will help position the contribution.
We appreciate your insightful suggestion. In response, we have revised the manuscript to better clarify how our study differs from earlier p53 structural investigations and hotspot analyses. These changes are incorporated at the end of the introduction and highlighted in red.
#3 For frequency and context distributions (e.g., core vs interface, DBD vs other domains), include hypothesis tests and effect sizes; report CIs for proportions (e.g., 54.3% at interfaces in AF model).
We appreciate the reviewer’s valuable comment. We have mapped the mutations onto 277 experimentally solved 3D structures of TP53. For example, the DNA-binding domain (DBD) is resolved as a monomer in some structures, while in others, it is solved as a homo-trimer or higher-order assemblies. Since no complete TP53 structure has been experimentally resolved, we utilised the AlphaFold (AF) model to provide an overall approximation of mutation distribution, particularly to assess how many mutations occur in core regions, interfaces, and other structural contexts. To support this, we have included a CSV file in the supplementary materials, which contains detailed characterisation data for all 277 structures. The file can be sorted by mutation frequency to identify hotspot mutations and their structural locations, including cases where mutations are found at protein–protein interfaces. In structures solved as higher-order assemblies, these interface mutations often occur between dimers.
#4 Discuss pLDDT/PAE thresholds and potential pitfalls when inferring interfaces/ligand sites from AF models; consider sensitivity analyses excluding low-confidence segments.
We appreciate the reviewer’s valuable comment. The reason for including the model is to see the distribution of reported mutations in full-length tp53. We have seen that DISOPRED3 Fig1B align well with low low-confidence score region from the AF model.
#5 Add ΔΔG units (kcal/mol) and clarify sign directions for each method; explain why FoldX values can be positive for destabilization and how “pathogenicity” (AlphaMissense) is thresholded. Consider adding a color bar for quick scanning.
We appreciate the reviewer’s valuable comment; I have added a section in the Materials and Methods section highlighted in red explain the sign for each tool
In FoldX, the formula is:
ΔΔG=ΔGmutant−ΔGwild
- ΔG = Gibbs free energy of folding (negative ΔG → stable protein)
- If the mutant has higher ΔG than wild-type, ΔΔG > 0 → destabilizing
- If the mutant has lower ΔG, ΔΔG < 0 → stabilizing
So, unlike Maestro, where negative ΔΔG means stabilizing, FoldX flips the interpretation.
#6 For each structure context (8DC8 monomer, 8R1G hetero-hexamer, 7EEU DNA-bound), summarize which residues switch categories (e.g., non-interface → interface) and how that might affect function—ideally with example structures highlighted.
We sincerely appreciate the reviewer’s valuable comment. In the second line of the Results section, we stated: “These structures encompass mutations at 368 of the 393 amino acid residues, representing approximately 93.6% of the protein sequence.” We initially attempted to colour-map these mutations on the 3D structure; however, since nearly all amino acid residues have reported mutations, the figure would not provide meaningful visual insights.
Moreover, the mapping of mutations is highly context-dependent. For example, in structure 8DC8, which is solved with a bound ligand, mutations around the ligand are classified as occurring “near the ligand-binding site”, whereas in structure 7EEU, the same residues are instead located at the protein–protein interface. To clarify this point, we have highlighted a section in the Results (marked in red) explaining how these context-specific differences may influence the protein’s functional behaviour.
#7 Expand on druggability: besides Y220C, briefly note interface- or zinc-site rescue strategies and how your atlas can prioritize alleles for small-molecule screening.
We thank the reviewer for this insightful suggestion. In the revised discussion, we have expanded our section on druggability to emphasise not only the well-characterised Y220C pocket, which has been successfully targeted by several small-molecule stabilisers, but also alternative strategies that exploit interface regions and zinc-binding sites. Several studies have demonstrated that small molecules can rescue mutant TP53 function by stabilising protein-protein interfaces or restoring zinc coordination in zinc-deficient variants. Our structural atlas, by mapping mutations across these critical regions, provides a framework to prioritise alleles most likely to benefit from such therapeutic interventions. This enables the rational selection of candidates for small-molecule screening and supports the development of precision strategies aimed at restoring TP53 function in chemoresistant tumours.
I sincerely appreciate the reviewer’s suggestions and recommendations, which have significantly enhanced the quality of the manuscript. Thank you once again for your valuable input.

Reviewer 3 Report
Comments and Suggestions for Authors
The manuscript presents a compelling and well-conceived computational study that systematically analyzes TP53 mutations from a structural perspective. The work has the potential to be a valuable resource for the cancer research community, providing a comprehensive map of how mutations impact p53 function. However, several critical issues need to be addressed before it is suitable for publication.
-
The manuscript title, "Mutational Disruption of TP53: A Structural Approach to Understanding Chemoresistance", suggests a strong link to practical applications. However, the work lacks a direct and detailed discussion on how these structural insights translate into drug resistance mechanisms. The authors must elaborate on potential targeted therapeutic strategies and cite relevant ongoing research in this area to provide practical conclusions.
-
The figures, which are central to the manuscript's findings, are of unacceptably low quality. Figure 1B, Figure 3, and Figure 4 are entirely illegible due to excessively small font sizes and poor resolution. The authors must provide high-quality, clear figures that are easy to read and interpret.
-
The manuscript contains numerous formatting errors. Citation style is inconsistent (e.g., "[1,2]" vs. "[8], [9]"), and in some cases, there are no spaces before citation brackets. The figure captions also contain errors (e.g., "(A.1)" should be "(A)" for Figure 1 and "(B1)" should be "(B)" for Figure 2).
-
The discussion section contains a grammatical error, as the second paragraph begins with a lowercase letter. The entire manuscript requires a thorough proofread to correct spelling, punctuation, and other minor mistakes that detract from its professionalism.
-
As a purely computational study, the findings are predictions. While not a prerequisite for all computational work, the authors should acknowledge this limitation and suggest future avenues for experimental validation to confirm their structural findings.
To sum up, the manuscript is a methodologically sound and potentially valuable study that explores a highly relevant topic in cancer biology. However, its current form is not suitable for publication. The authors must provide high-quality figures, correct all formatting and language errors, and, most importantly, significantly expand the discussion to directly connect their structural findings to practical applications and the issue of chemoresistance. The manuscript can be reconsidered for publication after a comprehensive revision addressing these critical issues.
Author Response
Reviwer3
The manuscript title, "Mutational Disruption of TP53: A Structural Approach to Understanding Chemoresistance", suggests a strong link to practical applications. However, the work lacks a direct and detailed discussion on how these structural insights translate into drug resistance mechanisms. The authors must elaborate on potential targeted therapeutic strategies and cite relevant ongoing research in this area to provide practical conclusions.
I appreciate the reviewer's time and effort in re-evaluating the paper, as well as the valuable suggestions and questions
The figures, which are central to the manuscript's findings, are of unacceptably low quality. Figure 1B, Figure 3, and Figure 4 are entirely illegible due to excessively small font sizes and poor resolution. The authors must provide high-quality, clear figures that are easy to read and interpret.
We thank the reviewer for this valuable feedback. We have carefully re-checked all the figures and confirm that they meet the journal’s resolution requirements and are provided in high-quality format. However, we understand the concern regarding readability, especially for Figures 1B, 3, and 4. To address this, we have increased the figure quality and font, which enhanced figure clarity, and ensured that the updated figures are easier to read and interpret even without zooming. The revised high-resolution figures have now been uploaded with the manuscript.
The manuscript contains numerous formatting errors. Citation style is inconsistent (e.g., "[1,2]" vs. "[8], [9]"), and in some cases, there are no spaces before citation brackets. The figure captions also contain errors (e.g., "(A.1)" should be "(A)" for Figure 1 and "(B1)" should be "(B)" for Figure 2).
We appreciate the reviewer’s valuable feedback. All cited references have been standardised, and the missing figure labels have been corrected.
The discussion section contains a grammatical error, as the second paragraph begins with a lowercase letter. The entire manuscript requires a thorough proofread to correct spelling, punctuation, and other minor mistakes that detract from its professionalism.
Thank you for the reviewer’s constructive comments. The discussion has been completely rewritten, and all corrections have been made and highlighted in red.
As a purely computational study, the findings are predictions. While not a prerequisite for all computational work, the authors should acknowledge this limitation and suggest future avenues for experimental validation to confirm their structural findings.
We appreciate the reviewer’s insightful comment and have included a short note at the end of the discussion highlighting the computational scope of this study and the necessity for experimental validation.
To sum up, the manuscript is a methodologically sound and potentially valuable study that explores a highly relevant topic in cancer biology. However, its current form is not suitable for publication. The authors must provide high-quality figures, correct all formatting and language errors, and, most importantly, significantly expand the discussion to directly connect their structural findings to practical applications and the issue of chemoresistance. The manuscript can be reconsidered for publication after a comprehensive revision addressing these critical issues.
I sincerely appreciate the reviewer’s suggestions and recommendations, which have significantly enhanced the quality of the manuscript. Thank you once again for your valuable input.

Round 2
Reviewer 3 Report
Comments and Suggestions for Authors
Thank you for providing the revised manuscript.
The Author has made a significant effort to address the majority of the points raised in my previous review. I appreciate the improvements made to the figures, as well as the expanded discussion on the implications of the findings.
However, I have noticed that the Author did not thoroughly proofread the text for formatting and citation style. The citation formatting remains inconsistent, with issues such as [19,20], [21] and missing spaces before some brackets (e.g., [1,2], [15], [26], [38]). The citation format [33], [34], [35], [36], [37] is also incorrect and inconsistent with the rest of the manuscript.
Therefore, while the scientific content is now suitable for publication, I recommend that the manuscript undergo a final, careful review by the Author and the Editorial Team to ensure all these minor formatting and citation errors are corrected.